# Asynchronous Anytime Sequential Monte Carlo

**Brooks Paige     Frank Wood**
Department of Engineering Science
University of Oxford
Oxford, UK
{brooks,fwood}@robots.ox.ac.uk

**Arnaud Doucet     Yee Whye Teh**
Department of Statistics
University of Oxford
Oxford, UK
{doucet,y.w.teh}@stats.ox.ac.uk

## Abstract

We introduce a new sequential Monte Carlo algorithm we call the *particle cascade*. The particle cascade is an asynchronous, anytime alternative to traditional sequential Monte Carlo algorithms that is amenable to parallel and distributed implementations. It uses no barrier synchronizations which leads to improved particle throughput and memory efficiency. It is an anytime algorithm in the sense that it can be run forever to emit an unbounded number of particles while keeping within a fixed memory budget. We prove that the particle cascade provides an unbiased marginal likelihood estimator which can be straightforwardly plugged into existing pseudo-marginal methods.

## 1   Introduction

Sequential Monte Carlo (SMC) inference techniques require blocking barrier synchronizations at resampling steps which limit parallel throughput and are costly in terms of memory. We introduce a new asynchronous anytime sequential Monte Carlo algorithm that has statistical efficiency competitive with standard SMC algorithms and has sufficiently higher particle throughput such that it is on balance more efficient per unit computation time. Our approach uses locally-computed decision rules for each particle that do not require block synchronization of all particles, instead only sharing of summary statistics with particles that follow. In our algorithm each resampling point acts as a queue rather than a barrier: each particle chooses the number of its own offspring by comparing its own weight to the weights of particles which previously reached the queue, blocking only to update summary statistics before proceeding.

An anytime algorithm is an algorithm that can be run continuously, generating progressively better solutions when afforded additional computation time. Traditional particle-based inference algorithms are not anytime in nature; all particles need to be propagated in lock-step to completion in order to compute expectations. Once a particle set runs to termination, inference cannot straightforwardly be continued by simply doing more computation. The naïve strategy of running SMC again and merging the resulting sets of particles is suboptimal due to bias (see [12] for explanation). Particle Markov chain Monte Carlo methods (i.e. particle Metropolis Hastings and iterated conditional sequential Monte Carlo (iCSMC) [1]) for correctly merging particle sets produced by additional SMC runs are closer to anytime in nature but suffer from burstiness as big sets of particles are computed then emitted at once and, fundamentally, the inner-SMC loop of such algorithms still suffers the kind of excessive synchronization performance penalty that the particle cascade directly avoids. Our asynchronous SMC algorithm, the *particle cascade*, is anytime in nature. The particle cascade can be run indefinitely, without resorting to merging of particle sets.

### 1.1   Related work

Our algorithm shares a superficial similarity to Bernoulli branching numbers [5] and other search and exploration methods used for particle filtering, where each particle samples some number of

children to propagate to the next observation. Like the particle cascade, the total number of particles which exist at each generation is allowed to gradually increase and decrease. However, computing branching correction numbers is generally a synchronous operation, requiring all particle weights to be known in order to choose an appropriate number of offspring; nor are these methods anytime. Sequentially interacting Markov chain Monte Carlo [2] is an anytime algorithm, which although conceptually similar to SMC has different synchronization properties.

Parallelizing the resampling step of sequential Monte Carlo methods has drawn increasing recent interest as the effort progresses to scale up algorithms to take advantage of high-performance computing systems and GPUs. Removing the global collective resampling operation [9] is a particular focus for improving performance.

Running arbitrarily many particles within a fixed memory budget can also be addressed by tracking random number seeds used to generate proposals, allowing particular particles to be deterministically "replayed" [7]. However, this approach is not asynchronous nor anytime.

## 2  Background

We begin by briefly reviewing sequential Monte Carlo as generally formulated on state-space models. Suppose we have a non-Markovian dynamical system with latent random variables $X_0, \ldots, X_N$ and observed random variables $Y_0, \ldots, Y_N$ described by the joint density

$$p(x_n|x_{0:n-1}, y_{0:n-1}) = f(x_n|x_{0:n-1})$$
$$p(y_n|x_{0:n}, y_{0:n-1}) = g(y_n|x_{0:n}), \tag{1}$$

where $X_0$ is drawn from some initial distribution $\mu(\cdot)$, and $f$ and $g$ are conditional densities.

Given observed values $Y_{0:N} = y_{0:N}$, the posterior distribution $p(x_{0:n}|y_{0:n})$ is approximated by a weighted set of $K$ particles, with each particle $k$ denoted $X_{0:n}^k$ for $k = 1, \ldots, K$. Particles are propagated forward from proposal densities $q(x_n|x_{0:n-1})$ and re-weighted at each $n = 1, \ldots, N$

$$X_n^k|X_{0:n-1}^k \sim q(x_n|X_{0:n-1}^k) \tag{2}$$

$$w_n^k = \frac{g(y_n|X_{0:n}^k)f(X_n^k|X_{0:n-1}^k)}{q(X_n^k|X_{0:n-1}^k)} \tag{3}$$

$$W_n^k = W_{n-1}^k w_n^k, \tag{4}$$

where $w_n^k$ is the weight associated with observation $y_n$ and $W_n^k$ is the unnormalized weight of particle $k$ after observation $n$. It is assumed that exact evaluation of $p(x_{0:N}|y_{0:N})$ is intractable and that the likelihoods $g(y_n|X_{0:n}^k)$ can be evaluated pointwise. In many complex dynamical systems, or in black-box simulation models, evaluation of $f(X_n^k|X_{0:n-1}^k)$ may be prohibitively costly or even impossible. As long as one is capable of simulating from the system, the proposal distribution can be chosen as $q(\cdot) \equiv f(\cdot)$, in which case the particle weights are simply $w_n^k = g(y_n|X_{0:n}^k)$, eliminating the need to compute the densities $f(\cdot)$.

The normalized particle weights $\bar{\omega}_n^k = W_n^k / \sum_{j=1}^K W_n^j$ are used to approximate the posterior

$$\hat{p}(x_{0:n}|y_{0:n}) \approx \sum_{k=1}^K \bar{\omega}_n^k \delta_{X_{0:n}^k}(x_{0:n}). \tag{5}$$

In the very simple sequential importance sampling setup described here, the marginal likelihood can be estimated by $\hat{p}(y_{0:n}) = \frac{1}{K} \sum_{k=1}^K W_n^k$.

### 2.1  Resampling and degeneracy

The algorithm described above suffers from a degeneracy problem wherein most of the normalized weights $\bar{\omega}_n^1, \ldots, \bar{\omega}_n^K$ become very close to zero for even moderately large $n$. Traditionally this is combated by introducing a resampling step: as we progress from $n$ to $n+1$, particles with high weights are duplicated and particles with low weights are discarded, preventing all the probability mass in our approximation to the posterior from accumulating on a single particle. A resampling

scheme is an algorithm for selecting the number of offspring particles $M_{n+1}^k$ that each particle $k$ will produce after stage $n$. Many different schemes for resampling particles exist; see [6] for an overview. Resampling changes the weights of particles: as the system progresses from $n$ to $n+1$, each of the $M_{n+1}^k$ children are assigned a new weight $V_{n+1}^k$, replacing the previous weight $W_n^k$ prior to resampling. Most resampling schemes generate an unweighted set of particles with $V_{n+1}^k = 1$ for all particles. When a resampling step is added at every $n$, the marginal likelihood can be estimated by $\hat{p}(y_{0:n}) = \prod_{i=0}^n \frac{1}{K} \sum_{k=1}^K w_i^k$; this estimate of the marginal likelihood is unbiased [8].

## 2.2 Synchronization and limitations

Our goal is to scale up to very large numbers of particles, using a parallel computing architecture where each particle is simulated as a separate process or thread. In order to resample at each $n$ we must compute the normalized weights $\bar{\omega}_n^k$, requiring us to wait until all individual particles have both finished forward simulation and computed their individual weight $W_n^k$ before the normalization and resampling required for any to proceed. While the forward simulation itself is trivially parallelizable, the weight normalization and resampling step is a synchronous, collective operation. In practice this can lead to significant underuse of computing resources in a multiprocessor environment, hindering our ability to scale up to large numbers of particles.

Memory limitations on finite computing hardware also limit the number of simultaneous particles we are capable of running in practice. All particles must move through the system together, simultaneously; if the total memory requirements of particles is greater than the available system RAM, then a substantial overhead will be incurred from swapping memory contents to disk.

# 3 The Particle Cascade

The particle cascade algorithm we introduce addresses both these limitations: it does not require synchronization, and keeps only a bounded number of particles alive in the system at any given time. Instead of resampling, we will consider particle branching, where each particle may produce 0 or more offspring. These branching events happen asynchronously and mutually exclusively, i.e. they are processed one at a time.

## 3.1 Local branching decisions

At each stage $n$ of sequential Monte Carlo, particles process observation $y_n$. Without loss of generality, we can define an ordering on the particles $1, 2, \ldots$ in the order they arrive at $y_n$. We keep track of the running average weight $\overline{W}_n^k$ of the first $k$ particles to arrive at observation $y_n$ in an online manner

$$\overline{W}_n^k = W_n^k \qquad\qquad \text{for } k = 1, \tag{6}$$

$$\overline{W}_n^k = \frac{k-1}{k}\overline{W}_n^{k-1} + \frac{1}{k}W_n^k \qquad\qquad \text{for } k = 2, 3, \ldots. \tag{7}$$

The number of children of particle $k$ depends on the weight $W_n^k$ of particle $k$ relative to those of other particles. Particles with higher relative weight are more likely to be located in a high posterior probability part of the space, and should be allowed to spawn more child particles.

In our online asynchronous particle system we do not have access to the weights of future particles when processing particle $k$. Instead we will compare $W_n^k$ to the current average weight $\overline{W}_n^k$ among particles processed thus far. Specifically, the number of children, which we denote by $M_{n+1}^k$, will depend on the ratio

$$R_n^k = \frac{W_n^k}{\overline{W}_n^k}. \tag{8}$$

Each child of particle $k$ will be assigned a weight $V_{n+1}^k$ such that the total weight of all children $M_{n+1}^k V_{n+1}^k$ has expectation $W_n^k$.

There is a great deal of flexibility available in designing a scheme for choosing the number of child particles; we need only be careful to set $V_{n+1}^k$ appropriately. Informally, we would like $M_{n+1}^k$ to

be large when $R_n^k$ is large. If $M_{n+1}^k$ is sampled in such a way that $\mathbb{E}[M_{n+1}^k] = R_n^k$, then we set the outgoing weight $V_{n+1}^k = \overline{W}_n^k$. Alternatively, if we are using a scheme which deterministically guarantees $M_{n+1}^k > 0$, then we set $V_{n+1}^k = W_n^k / M_{n+1}^k$.

A simple approach would be to sample $M_{n+1}^k$ independently conditioned on the weights. In such schemes we could draw each $M_{n+1}^k$ from some simple distribution, e.g. a Poisson distribution with mean $R_n^k$, or a discrete distribution over the integers $\{\lfloor R_n^k \rfloor, \lceil R_n^k \rceil\}$. However, one issue that arises in such approaches where the number of children for each particle is conditionally independent is that the variance of the total number of particles at each generation can grow faster than desirable. Suppose we start the system with $K_0$ particles. The number of particles at subsequent stages $n$ is given recursively as $K_n = \sum_{k=1}^{K_{n-1}} M_n^k$. We would like to avoid situations in which the number of particles becomes too large, or collapses to 1.

Instead, we will allow $M_n^k$ to depend on the number of children of previous particles at $n$, in such a way that we can stabilize the total number of particles in each generation. Suppose that we wish for the number of particles to be stabilized around $K_0$. After $k-1$ particles have been processed, we expect the total number of children produced at that point to be approximately $k-1$, so that if the number is less than $k-1$ we should allow particle $k$ to produce more children, and vice versa. Similarly, if we already currently have more than $K_0$ children, we should allow particle $k$ to produce fewer children.

We use a simple scheme which satisfies these criteria, where the number of particles is chosen at random when $R_n^k < 1$, and set deterministically when $R_n^k \geq 1$

$$(M_{n+1}^k, V_{n+1}^k) = \begin{cases} (0,0) \text{ w.p. } 1 - R_n^k, & \text{if } R_n^k < 1; \\ (1, \overline{W}_n^k) \text{ w.p. } R_n^k, & \text{if } R_n^k < 1; \\ \left(\lfloor R_n^k \rfloor, \frac{W_n^k}{\lfloor R_n^k \rfloor}\right) & \text{if } R_n^k \geq 1 \text{ and } \sum_{j=1}^{k-1} M_{n+1}^j > \min(K_0, k-1); \\ \left(\lceil R_n^k \rceil, \frac{W_n^k}{\lceil R_n^k \rceil}\right) & \text{if } R_n^k \geq 1 \text{ and } \sum_{j=1}^{k-1} M_{n+1}^j \leq \min(K_0, k-1). \end{cases} \quad (9)$$

As the number of particles becomes large, the estimated average weight closely approximates the true average weight. Were we to replace the deterministic rounding with a Bernoulli$(R_n^k - \lfloor R_n^k \rfloor)$ choice between $\{\lfloor R_n^k \rfloor, \lceil R_n^k \rceil\}$, then this decision rule defines the same distribution on the number of offspring particles $M_{n+1}^k$ as the well-known systematic resampling procedure [3, 9].

Note the anytime nature of this algorithm — any given particle passing through the system needs only the running average $\overline{W}_n^k$ and the preceding child particle counts $\sum_{j=1}^{k-1} M_{n+1}^j$ in order to make local branching decisions, not the previous particles themselves. Thus it is possible to run this algorithm for some fixed number of initial particles $K_0$, inspect the output of the completed particles which have left the system, and decide whether to continue by initializing additional particles.

## 3.2 Computing expectations and marginal likelihoods

Samples drawn from the particle cascade can be used to compute expectations in the same manner as usual; that is, given some function $\varphi(\cdot)$, we normalize weights $\bar{\omega}_n^k = W_n^k / \sum_{j=1}^{K_n} W_n^j$ and approximate the posterior expectation by $\mathbb{E}[\varphi(X_{0:n})|y_{0:n}] \approx \sum_{k=1}^{K_n} \bar{\omega}_n^k \varphi(X_{0:n}^k)$.

We can also use the particle cascade to define an estimator of the marginal likelihood $p(y_{0:n})$,

$$\hat{p}(y_{0:n}) = \frac{1}{K_0} \sum_{k=1}^{K_n} W_n^k. \quad (10)$$

The form of this estimate is fairly distinct from the standard SMC estimators in Section 2. One can think of $\hat{p}(y_{0:n})$ as $\hat{p}(y_{0:n}) = \hat{p}(y_0) \prod_{i=1}^n \hat{p}(y_i|y_{0:i-1})$ where

$$\hat{p}(y_0) = \frac{1}{K_0} \sum_{k=1}^{K_0} W_0^k, \qquad \hat{p}(y_n|y_{0:n-1}) = \frac{\sum_{k=1}^{K_n} W_n^k}{\sum_{k=1}^{K_{n-1}} W_{n-1}^k} \text{ for } n \geq 1. \quad (11)$$

Note that the incrementally updated running averages $\overline{W}_n^k$ are very directly tied to the marginal likelihood estimate; that is, $\hat{p}(y_{0:n}) = \frac{K_n}{K_0} \overline{W}_n^k$.

### 3.3 Theoretical properties, unbiasedness, and consistency

Under weak assumptions we can show that the marginal likelihood estimator $\hat{p}(y_{0:n})$ defined in Eq. 10 is unbiased, and that both its variance and L2 errors of estimates of reasonable posterior expectations decrease in the number of particle initializations as $1/K_0$. Note that because the cascade is an anytime algorithm $K_0$ may be increased simply, without restarting inference. Detailed proofs are given in the supplemental material; statements of the results are provided here.

Denote by $B(E)$ the space of bounded real-valued functions on a space $E$, and suppose each $X_n$ is an $\mathcal{X}$-valued random variable. Assume the Bernoulli$(R_n^k - \lfloor R_n^k \rfloor)$ version of the resampling rule in Eq. 9, and further assume that $g(y_n|\cdot, y_{0:n-1}) : \mathcal{X}^{n+1} \to \mathbb{R}$ is in $B(\mathcal{X}^{n+1})$ and strictly positive. Finally assume that the ordering in which particles arrive at each $n$ is a random permutation of the particle index set, conditions which we state precisely in the supplemental material. Then the following propositions hold:

**Proposition 1 (Unbiasedness of marginal likelihood estimate)** *For any $K_0 \geq 1$ and $n \geq 0$*

$$\mathbb{E}\left[\hat{p}(y_{0:n})\right] = p(y_{0:n}). \tag{12}$$

**Proposition 2 (Variance of marginal likelihood estimate)** *For any $n \geq 0$, there exists a constant $a_n < \infty$ such that for any $K_0 \geq 1$*

$$\mathbb{V}\left[\hat{p}(y_{0:n})\right] \leq \frac{a_n}{K_0}. \tag{13}$$

**Proposition 3 (L2 error bounds)** *For any $n \geq 0$, there exists a constant $\bar{a}_n < \infty$ such that for any $K_0 \geq 1$ and any $\psi_n \in B\left(\mathcal{X}^{n+1}\right)$*

$$\mathbb{E}\left[\left\{\left(\sum_{k=1}^{K_n} \bar{\omega}_n^k \psi_n(X_{0:n}^k)\right) - \int p(dx_{0:n}|y_{0:n})\psi_n(x_{0:n})\right\}^2\right] \leq \frac{\bar{a}_n}{K_0}\left\|\psi_n\right\|^2. \tag{14}$$

Additional results and proofs can be found in the supplemental material.

## 4 Active bounding of memory usage

In an idealized computational environment, with infinite available memory, our implementation of the particle cascade could begin by launching (a very large number) $K_0$ particles simultaneously which then gradually propagate forward through the system. In practice, only some finite number of particles, probably much smaller than $K_0$, can be simultaneously simulated efficiently. Furthermore, the initial particles are not truly launched all at once, but rather in a sequence, introducing a dependency in the order in which particles arrive at each observation $n$.

Our implementation of the particle cascade addresses these issues by explicitly injecting randomness into the execution order of particles, and by imposing a machine-dependent hard cap on the number of simultaneous extant processes. This permits us to run our particle filter system indefinitely, for arbitrarily large and, in fact, growing initial particle counts $K_0$, on fixed commodity hardware.

Each particle in our implementation runs as an independent operating system process [11]. In order to efficiently run a large number of particles, we impose a hard limit $\rho$ on the total number of particles which can simultaneously exist in the particle system; most of these will generally be sleeping processes. The ideal choice for this number will vary based on hardware capabilities, but in general should be made as large as possible.

Scheduling across particles is managed via a global *first-in random-out* process queue of length $\rho$; this can equivalently be conceptualized as a random-weight priority queue. Each particle corresponds to a single live process, augmented by a single additional control process which is responsible only for spawning additional initial particles (i.e. incrementing the initial particle count $K_0$). When any particle $k$ arrives at any likelihood evaluation $n$, it computes its target number of child particles $M_{n+1}^k$ and outgoing particle weight $V_{n+1}^k$. If $M_{n+1}^k = 0$ it immediately terminates; otherwise it enters the queue. Once this particle either enters the queue or terminates, some other process

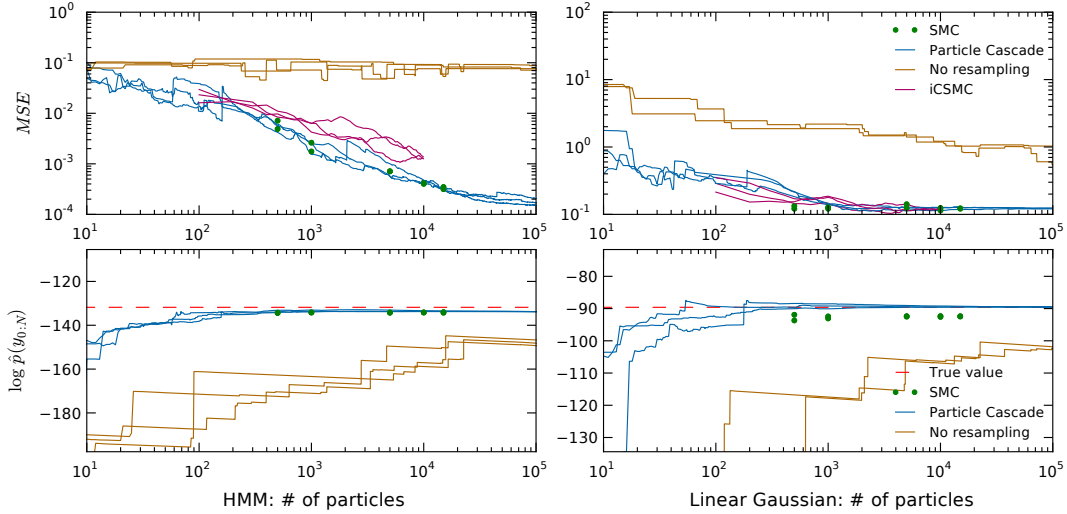

Figure 1: All results are reported over multiple independent replications, shown here as independent lines. (top) Convergence of estimates to ground truth vs. number of particles, shown as (left) MSE of marginal probabilities of being in each state for every observation $n$ in the HMM, and (right) MSE of the latent expected position in the linear Gaussian state space model. (bottom) Convergence of marginal likelihood estimates to the ground truth value (marked by a red dashed line), for (left) the HMM, and (right) the linear Gaussian model.

continues execution — this process is chosen uniformly at random, and as such may be a sleeping particle at any stage $n < N$, or it may instead be the control process which then launches a new particle. At any given time, there are some number of particles $K_\rho < \rho$ currently in the queue, and so the probability of resuming any particular individual particle, or of launching a new particle, is $1/(K_\rho + 1)$. If the particle released from the queue has exactly one child to spawn, it advances to the next observation and repeats the resampling process. If, however, a particle has more than one child particle to spawn, rather than launching all child particles at once it launches a single particle to simulate forward, decrements the total number of particles left to launch by one, and itself re-enters the queue. The system is initialized by seeding the system with a number of initial particles $\rho_0 < \rho$ at $n = 0$, creating $\rho_0$ active initial processes. The ideal choice for the process count constraint $\rho$ may vary across operating systems and hardware.

In the event that the process count is fully saturated (i.e. the process queue is full), then we forcibly prevent particles from duplicating themselves and creating new children. If we release a particle from the queue which seeks to launch $m > 1$ additional particles when the queue is full, we instead collapse all the remaining particles into a single particle; this single particle represents a virtual set of particles, but does not create a new process and requires no additional CPU or memory resources. We keep track of a particle count multiplier $C_n^k$ that we propagate forward along with the particle. All particles are initialized with $C_0^k = 1$, and then when a particle collapse takes place, update their multiplier at $n + 1$ to $mC_n^k$. This affects the way in which running weight averages are computed; suppose a new particle $k$ arrives with multiplier $C_n^k$ and weight $W_n^k$. We incorporate all these values into the average weight immediately, and update $\overline{W}_n^k$ taking into account the multiplicity, with

$$\overline{W}_n^k = \frac{k-1}{k + C_n^k - 1}\overline{W}_n^{k-1} + \frac{C_n^k}{k + C_n^k - 1}W_n^k \qquad \text{for } k = 2, 3, \ldots. \qquad (15)$$

This does not affect the computation of the ratio $R_n^k$. We preserve the particle multiplier, until we reach the final $n = N$; then, after all forward simulation is complete, we re-incorporate the particle multiplicity when reporting the final particle weight $W_N^k = C_N^k V_N^k w_N^k$.

## 5    Experiments

We report experiments on performing inference in two simple state space models, each with $N = 50$ observations, in order to demonstrate the overall validity and utility of the particle cascade algorithm.

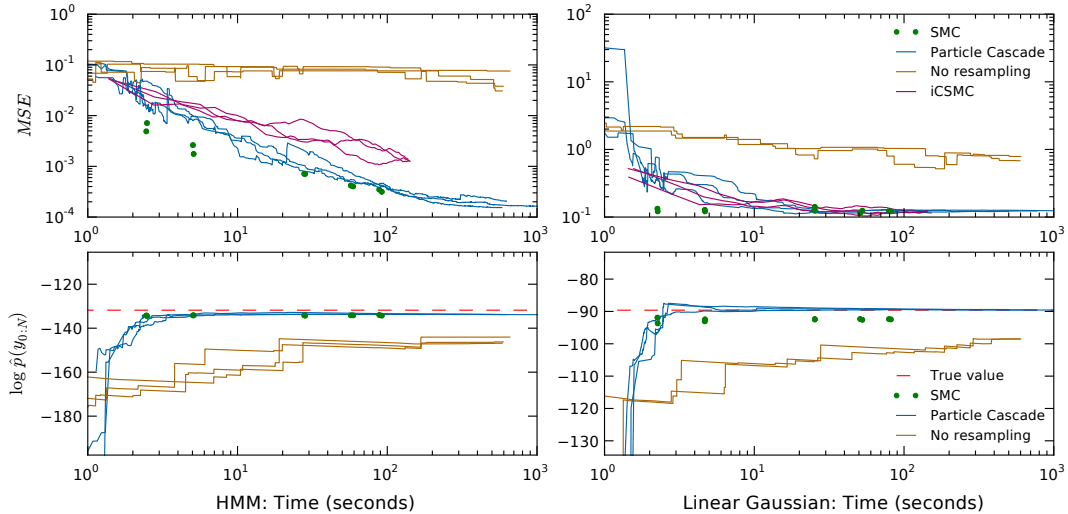

Figure 2: (top) Comparative convergence rates between SMC alternatives including our new algorithm, and (bottom) estimation of marginal likelihood, by time. Results are shown for (left) the hidden Markov model, and (right) the linear Gaussian state space model.

The first is a hidden Markov model (HMM) with 10 latent discrete states, each with an associated Gaussian emission distribution; the second a one-dimensional linear Gaussian model. Note that using these models means that we can compute posterior marginals at each $n$ and the marginal likelihood $Z = p(y_{0:N})$ exactly.

These experiments are not designed to stress-test the particle cascade; rather, they are designed to show that performance of the particle cascade closely approximates that of fully synchronous SMC algorithms, even in a small-data small-complexity regime where we expect their performance to be very good. In addition to comparing to standard SMC, we also compare to a worst-case particle filter in which we never resample, instead propagating particles forward deterministically with a single child particle at every $n$. While the statistical (per-sample) efficiency of this approach is quite poor, it is fully parallelizable with no blocking operations in the algorithm at all, and thus provides a ceiling estimate of the raw sampling speed attainable in our overall implementation.

We also benchmark against what we believe to be the most practically competitive similar approach, iterated conditional SMC [1]. Iterated

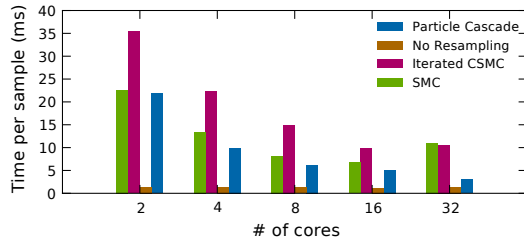

Figure 3: Average time to draw a single complete particle on a variety of machine architectures. Queueing rather than blocking at each observation improves performance, and appears to improve relative performance even more as the available compute resources increase. Note that this plot shows only average time per sample, not a measure of statistical efficiency. The high speed of the non-resampling algorithm is not sufficient to make it competitive with the other approaches.

conditional SMC corresponds to the particle Gibbs algorithm in the case where parameter values are known; by using a particle filter sweep as a step within a larger MCMC algorithm, iCSMC provides a statistically valid approach to sampling from a posterior distribution by repeatedly running sequential Monte Carlo sweeps each with a fixed number of particles. One downside to iCSMC is that it does not provide an estimate of the marginal likelihood. In all benchmarks, we propose from the prior distribution, with $q(x_n|\cdot) \equiv f(x_n|x_{0:n-1})$; the SMC and iCSMC benchmarks use a multinomial resampling scheme.

On both these models we see the statistical efficiency of the particle cascade is approximately in line with synchronous SMC, slightly outperforming the iCSMC algorithm and significantly outperform-

ing the fully parallelized non-resampling approach. This suggests that the approximations made by computing weights at each $n$ based on only the previously observed particles, and the total particle count limit imposed by $\rho$, do not have an adverse effect on overall performance. In Fig. 1 we plot convergence per particle to the true posterior distribution, as well as convergence in our estimate of the normalizing constant.

## 5.1 Performance and scalability

Although values will be implementation-dependent, we are ultimately interested not in per-sample efficiency but rather in our rate of convergence over time. We record wall clock time for each algorithm for both of these models; the results for convergence of our estimates of values and marginal likelihood are shown in Fig. 2. These particular experiments were all run on Amazon EC2, in an 8-core environment with Intel Xeon E5-2680 v2 processors. The particle cascade provides a much faster and more accurate estimate of the marginal likelihood than the competing methods, in both models. Convergence in estimates of values is quick as well, faster than the iCSMC approach. We note that for very small numbers of particles, running a simple particle filter is faster than the particle cascade, despite the blocking nature of the resampling step. This is due to the overhead incurred by the particle cascade in sending an initial flurry of $\rho_0$ particles into the system before we see any particles progress to the end; this initial speed advantage diminishes as the number of samples increases. Furthermore, in stark contrast to the simple SMC method, there are no barriers to drawing more samples from the particle cascade indefinitely. On this fixed hardware environment, our implementation of SMC, which aggressively parallelizes all forward particle simulations, exhibits a dramatic loss of performance as the number of particles increases from $10^4$ to $10^5$, to the point where simultaneously running $10^5$ particles is simply not possible in a feasible amount of time.

We are also interested in how the particle cascade scales up to larger hardware, or down to smaller hardware. A comparison across five hardware configurations is shown in Fig. 3.

## 6 Discussion

The particle cascade has broad applicability to all SMC and particle filtering inference applications. For example, constructing an appropriate sequence of densities for SMC is possible in arbitrary probabilistic graphical models, including undirected graphical models; see e.g. the sequential decomposition approach of [10]. We are particularly motivated by the SMC-based probabilistic programming systems that have recently appeared in the literature [13, 11]. Both suggested that the primary performance bottleneck in their inference algorithms was barrier synchronization, something we have done away with entirely. What is more, while particle MCMC methods are particularly appropriate when there is a clear boundary that can be exploited between between parameters of interest and nuisance state variables, in probabilistic programming in particular, parameter values must be generated as part of the state trajectory itself, leaving no explicitly denominated latent parameter variables per se. The particle cascade is particularly relevant in such situations.

Finally, as the particle cascade yields an unbiased estimate of the marginal likelihood it can be plugged directly into PIMH, SMC$^2$ [4], and other existing pseudo-marginal methods.

### Acknowledgments

Yee Whye Teh's research leading to these results has received funding from EPSRC (grant EP/K009362/1) and the ERC under the EU's FP7 Programme (grant agreement no. 617411). Arnaud Doucet's research is partially funded by EPSRC (grants EP/K009850/1 and EP/K000276/1). Frank Wood is supported under DARPA PPAML through the U.S. AFRL under Cooperative Agreement number FA8750-14-2-0004. The U.S. Government is authorized to reproduce and distribute reprints for Governmental purposes notwithstanding any copyright notation heron. The views and conclusions contained herein are those of the authors and should be not interpreted as necessarily representing the official policies or endorsements, either expressed or implied, of DARPA, the U.S. Air Force Research Laboratory or the U.S. Government.

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
