[Supplementary Material]

# Asynchronous Anytime Sequential Monte Carlo: Supplemental Material

Brooks Paige      Frank Wood
Department of Engineering Science
University of Oxford
Oxford, UK
{brooks,fwood}@robots.ox.ac.uk

Arnaud Doucet      Yee Whye Teh
Department of Statistics
University of Oxford
Oxford, UK
{doucet,y.w.teh}@stats.ox.ac.uk

## 1   Introduction

We have the following state-space model

$$X_0 \sim \mu,$$
$$X_n | X_{0:n-1} = x_{0:n-1}, Y_{0:n-1} \sim f_n\left(x_n | x_{0:n-1}\right) \qquad \text{for } n \geq 1,$$
$$Y_n | X_{0:n} = x_{0:n}, Y_{0:n-1} \sim g_n\left(y_n | x_{0:n}, y_{0:n-1}\right) \qquad \text{for } n \geq 0$$

where $X_n$ is a $\mathcal{X}-$valued random variable and $\mathcal{X}$ a metric space. Given a realization of the observations $Y_{0:t} = y_{0:t}$, we are interested in making inference about the latent state variables. We introduce the following unnormalised measures [2] for any $0 \leq n \leq t$,

$$\alpha_n\left(dx_{0:n}\right) = p\left(dx_{0:n}, y_{0:n}\right), \ \widehat{\alpha}_{n+1}\left(dx_{0:n+1}\right) = p\left(dx_{0:n+1}, y_{0:n}\right).$$

with normalisation constant $p(y_{0:n})$ and their normalised versions

$$\eta_n\left(dx_{0:n}\right) = p\left(dx_{0:n} | y_{0:n}\right), \widehat{\eta}_{n+1}\left(dx_{0:n+1}\right) = p\left(dx_{0:n+1} | y_{0:n}\right).$$

If $\mu\left(dx\right)$ is a measure, $\psi\left(x\right)$ a real-valued function, $K\left(dx' | x\right)$ a Markov

kernel and $A$ a Borel set, we use the following standard notation

$$\mu\left(\psi\right) = \int \mu\left(dx\right)\psi\left(x\right),$$

$$\mu K\left(A\right) = \int_A \mu\left(dx\right)K\left(\left.dx'\right|x\right),$$

$$K\psi\left(x\right) = \int \psi\left(x'\right)K\left(\left.dx'\right|x\right).$$

Using this notation, we have

$$\alpha_n\left(\psi\right) = \widehat{\alpha}_n\left(g_n\psi\right), \ \widehat{\alpha}_{n+1}\left(\psi\right) = \alpha_n \ f_n\left(\psi\right),$$

$$\eta_n\left(\psi\right) = \frac{\widehat{\alpha}_n\left(g_n\psi\right)}{\widehat{\alpha}_n\left(g_n\right)}, \ \widehat{\eta}_{n+1}\left(\psi\right) = \eta_n \ f_n\left(\psi\right).$$

The following particle algorithm is used.

- Initialisation $n = 0$. For $i = 1, ..., N_0$ Sample $X_0^{i,0} \sim \mu\left(\cdot\right)$ and compute $W_0^i = g_0\left(\left.y_0\right|X_0^{i,0}\right)$.

- At time $n \geq 0$.

  - Branching step: Resample $\left\{W_n^i, X_{0:n}^{i,n}\right\}_{i=1}^{N_n}$ to obtain $\left\{\widetilde{W}_n^i, X_{0:n}^{i,n+1}\right\}_{i=1}^{N_{n+1}}$.

  - Extension step: For $i = 1, ..., N_{n+1}$ sample $X_{n+1}^{i,n+1} \sim f_{n+1}\left(\left.\cdot\right|X_{0:n}^{i,n+1}\right)$.

  - Reweighing step: Set $W_{n+1}^i = \widetilde{W}_n^i \cdot g_{n+1}\left(\left.y_{n+1}\right|X_{0:n+1}^{i,n+1}, y_{0:n}\right)$.

On the branching step, we assume that the particles are processed sequentially in order given by a permutation $\sigma_n$ on $[N_n]$. The $i$th particle processed is $\sigma_n(i)$, and the number of children $M_{n+1}^i$ and common weight of each child $V_n^i$ are determined, based only on information of particles $\sigma_n(1), \ldots, \sigma_n(i)$, but not later particles and satisfy

$$V_n^i = \overline{W}_n^i = \frac{1}{i}\sum_{j=1}^i W_n^{\sigma_n(j)},$$

$$M_{n+1}^i = \left\lfloor \frac{W_n^{\sigma_n(i)}}{\overline{W}_n^i} \right\rfloor + \text{Bernoulli}\left(\frac{W_n^{\sigma_n(i)}}{\overline{W}_n^i} - \left\lfloor \frac{W_n^{\sigma_n(i)}}{\overline{W}_n^i} \right\rfloor\right).$$

The total number of children for the next stage is $N_{n+1} = \sum_{i=1}^{N_n} M_{n+1}^i$, with weights $\left(\widetilde{W}_n^i\right)_{i=1}^{N_{n+1}} = (\underbrace{V_n^1, \ldots, V_n^1}_{M_{n+1}^1}, \ldots, \underbrace{V_n^{N_n}, \ldots, V_n^{N_n}}_{M_{n+1}^{N_n}}).$

At each time step, we have the following approximations $\beta_n^{N_0}$ and $\widetilde{\beta}_n^{N_0}$ of $\alpha_n$ and the approximation $\widehat{\beta}_{n+1}^{N_0}$ of $\widehat{\alpha}_{n+1}$ :

$$\beta_n^{N_0}(dx_{0:n}) = \frac{\sum_{i=1}^{N_n} W_n^i \delta_{X_{0:n}^{i,n+1}}(dx_{0:n})}{N_0}$$

$$\widetilde{\beta}_n^{N_0}(dx_{0:n}) = \frac{\sum_{i=1}^{N_{n+1}} \widetilde{W}_n^i \delta_{X_{0:n}^{i,n+1}}(dx_{0:n})}{N_0},$$

$$\widehat{\beta}_{n+1}^{N_0}(dx_{0:n+1}) = \frac{\sum_{i=1}^{N_{n+1}} \widetilde{W}_n^i \delta_{X_{0:n+1}^{i,n+1}}(dx_{0:n})}{N_0},$$

Practically, when performing state estimation, we are not interested in the un-normalised measures $\beta_n^{N_0}$, $\widetilde{\beta}_n^{N_0}$ and $\widehat{\beta}_{n+1}^{N_0}$ but in their normalised versions defined as

$$\nu_n^{N_0}(dx_{0:n}) = \frac{\beta_n^{N_0}(dx_{0:n})}{\beta_n^{N_0}(1)}, \ \widetilde{\nu}_n^{N_0}(dx_{0:n}) = \frac{\widetilde{\beta}_n^{N_0}(dx_{0:n})}{\widetilde{\beta}_n^{N_0}(1)},$$

$$\widehat{\nu}_{n+1}^{N_0}(dx_{0:n+1}) = \frac{\widehat{\beta}_{n+1}^{N_0}(dx_{0:n+1})}{\widehat{\beta}_{n+1}^{N_0}(1)},$$

where $\nu_n^{N_0}$ and $\widetilde{\nu}_n^{N_0}$ approximate $\eta_n$ while $\widehat{\nu}_{n+1}^{N_0}$ approximates $\widehat{\eta}_{n+1}$.

This particle filter also ouputs an estimate of the marginal likelihood given by

$$\widehat{p}^{N_0}(y_{0:n}) = \widehat{p}^{N_0}(y_0) \prod_{k=1}^{n} \widehat{p}^{N_0}(y_k | y_{0:k-1})$$

where $\widehat{p}^{N_0}(y_0) := \frac{1}{N_0} \sum_{i=1}^{N_0} W_0^i$ and for $k \geq 1$

$$\widehat{p}^{N_0}(y_k | y_{0:k-1}) := \int g_k(y_k | x_{0:k}, y_{0:k-1}) \widehat{\nu}_{k-1}^{N_0}(dx_{0:k})$$

$$= \frac{\sum_{i=1}^{N_k} W_k^i}{\sum_{i=1}^{N_{k-1}} W_{k-1}^i} \text{ for } k \geq 1.$$

Hence it follows that

$$\widehat{p}^{N_0}(y_{0:n}) = \frac{1}{N_0} \sum_{i=1}^{N_n} W_n^i. \tag{1}$$

We denote by $B(E)$ the space of bounded real-valued functions on a space $E$, equipped with the sup norm denoted $\|f\| = \sup_{x \in E} |f(x)|$. We also denote by $\mathcal{F}_n$ the natural filtration associated with all random variables generated by the particle algorithm at the end of the $n$th reweighting step, and $\widetilde{\mathcal{F}}_n$ similarly for just after the the branching step.

We make the following assumption on the model and branching step.

**Assumption B**. The function $g_n(y_n | \cdot, y_{0:n-1}) : \mathcal{X}^{n+1} \to \mathbb{R}$ satisfies $g_n(y_n | x_{0:n}, y_{0:n-1}) > 0$ for all $x_{0:n} \in \mathcal{X}^{n+1}$ and $\|g_n(y_n | \cdot, y_{0:n-1})\| \leq 1$ for all $n \geq 0$.

We note that if $\|g_n(y_n|\cdot, y_{0:n-1})\| \leq B_n$ for some known constant $B_n$, then we can simply rescale $g_n(y_n|\cdot, y_{0:n-1})$ to satisfy Assumption B. The assumption that $g_n(y_n|x_{0:n}, y_{0:n-1}) > 0$ for all $x_{0:n}$ is a sufficient assumption ensuring the system of particles cannot die.

**Assumption O**. The particle ordering $\sigma_n$ is independent of all other random variables generating $\mathcal{F}_n$, conditioned on the number of particles $N_n$, and $\sigma_n$ is uniformly distributed across all permutations of $\{1, \ldots, N_n\}$.

It is straightforward to establish that the particle branching mechanism implies that $\Pr(N_n > 0) = 1$ for any $n \geq 0$ and that the following unbiasedness property is satisfied for any $\psi \in B(\mathcal{X}^n)$

$$\mathbb{E}\left[\left.\sum_{i=1}^{N_{n+1}} \widetilde{W}_n^i \, \psi\left(X_{0:n}^{i,n+1}\right) \right| \mathcal{F}_n\right] = \sum_{i=1}^{N_n} W_n^i \, \psi\left(X_{0:n}^{i,n}\right). \tag{2}$$

Additionally, it ensures that for each $n$ and $i$, we have

$$\mathbb{V}[M_n^i|\mathcal{F}_n] \leq V = 1/4 \tag{3}$$

as $M_n^i$ is a shifted Bernoulli random variable and $W_n^i, \widetilde{W}_n^i \leq 1$ straightforwardly by induction as $\|g_n(y_n|\cdot, y_{0:n-1})\| \leq 1$.

In the rest of the paper, Assumption B and Assumption O are assumed to hold.

## 2 Marginal likelihood estimation and unbiasedness

In this Section, we established that the marginal likelihood estimate given in (1) is unbiased.

**Proposition 1** *For any $N_0 \geq 1$ and $n \geq 0$, we have*

$$\mathbb{E}\left[\widehat{p}^{N_0}(y_{0:n})\right] = p(y_{0:n}).$$

**Proof.** The proof follows from a backward induction. We have

$$\mathbb{E}\left[\widehat{p}^{N_0}\left(y_{0:n}\right)\right] = \mathbb{E}\left[\mathbb{E}\left[\left.\frac{1}{N_0}\sum_{i=1}^{N_n}W_n^i\right|\widetilde{\mathcal{F}}_{n-1}\right]\right]$$

$$= \mathbb{E}\left[\frac{1}{N_0}\sum_{i=1}^{N_n}\widetilde{W}_{n-1}^i\underbrace{\int f_n\left(x_n|X_{0:n-1}^{i,n}\right)g_n\left(y_n|X_{0:n-1}^{i,n},x_n,y_{0:n-1}\right)dx_n}_{p\left(y_n|X_{0:n-1}^{i,n},y_{0:n-1}\right)}\right] \quad \left(\text{using } W_n^i=\widetilde{W}_{n-1}^i\cdot g_n\left(\cdot\right)\right)$$

$$= \mathbb{E}\left[\mathbb{E}\left[\left.\frac{1}{N_0}\sum_{i=1}^{N_n}\widetilde{W}_{n-1}^ip\left(y_n|X_{0:n-1}^{i,n},y_{0:n-1}\right)\right|\mathcal{F}_{n-1}\right]\right] \quad \left(\text{using (2)}\right)$$

$$= \mathbb{E}\left[\mathbb{E}\left[\left.\frac{1}{N_0}\sum_{i=1}^{N_{n-1}}W_{n-1}^ip\left(y_n|X_{0:n-1}^{i,n-1},y_{0:n-1}\right)\right|\widetilde{\mathcal{F}}_{n-2}\right]\right]$$

$$= \mathbb{E}\left[\mathbb{E}\left[\left.\frac{1}{N_0}\sum_{i=1}^{N_{n-1}}\widetilde{W}_{n-2}^ip\left(y_{n-1:n}|X_{0:n-2}^{i,n-1},y_{0:n-2}\right)\right|\mathcal{F}_{n-2}\right]\right]$$

$$= \mathbb{E}\left[\mathbb{E}\left[\left.\frac{1}{N_0}\sum_{i=1}^{N_{n-2}}W_{n-2}^ip\left(y_{n-1:n}|X_{0:n-2}^{i,n-2},y_{0:n-2}\right)\right|\widetilde{\mathcal{F}}_{n-3}\right]\right]$$

$$= \mathbb{E}\left[\frac{1}{N_0}\sum_{i=1}^{N_0}W_0^ip\left(y_{1:n}|X_0^{i,0},y_0\right)\right]$$

$$= p\left(y_{0:n}\right).$$

∎

## 3   L2 Error Bounds

We first establish L2 error bounds for the unnormalised measures $\beta_n^{N_0}$, $\widetilde{\beta}_n^{N_0}$ and $\widehat{\beta}_n^{N_0}$.

**Theorem 2 (L2 error bounds for unnormalised measures)** *For any $n \geq 0$, there exists $a_n, b_n, c_n < \infty$ such that for any $N_0 \geq 1$ and any $\psi_n \in B\left(\mathcal{X}^{n+1}\right)$, $\psi_{n+1} \in B\left(\mathcal{X}^{n+2}\right)$*

$$\mathbb{E}\left[\left\{\beta_n^{N_0}\left(\psi_n\right)-\alpha_n\left(\psi_n\right)\right\}^2\right] \leq \frac{a_n}{N_0}\left\|\psi_n\right\|^2,$$

$$\mathbb{E}\left[\left\{\widetilde{\beta}_n^{N_0}\left(\psi_n\right)-\alpha_n\left(\psi_n\right)\right\}^2\right] \leq \frac{b_n}{N_0}\left\|\psi_n\right\|^2,$$

$$\mathbb{E}\left[\left\{\widehat{\beta}_{n+1}^{N_0}\left(\psi_{n+1}\right)-\widehat{\alpha}_{n+1}\left(\psi_{n+1}\right)\right\}^2\right] \leq \frac{c_n}{N_0}\left\|\psi_{n+1}\right\|^2.$$

Using the function $\psi_n(x_{0:n}) = 1$, we get control over the variance of the unbiased estimator for the marginal likelihood estimate.

**Corollary 3** *We have, for some constant $a_n$,*

$$\mathbb{V}\left[\frac{1}{N_0}\sum_{i=1}^{N_n}W_n^i\right] \leq \frac{a_n}{N_0}.$$

We proof this result by induction on $n$. It is straightforward to check that there exists $a_0 < \infty$ such that $\mathbb{E}\left[\left\{\beta_0^{N_0}\left(\psi_0\right)-\alpha_0\left(\psi_0\right)\right\}^2\right] \leq \frac{a_0}{N_0}\left\|\psi_0\right\|^2$ holds as the initial particles are i.i.d. The proof then relies on the following propositions.

**Proposition 4 (Branching Step)** *Assume that there exists $a_n < \infty$ such that for any $\psi_n \in B\left(\mathcal{X}^{n+1}\right)$*

$$\mathbb{E}\left[\left\{\beta_n^{N_0}\left(\psi_n\right) - \alpha_n\left(\psi_n\right)\right\}^2\right] \leq \frac{a_n}{N_0}\left\|\psi_n\right\|^2 \tag{4}$$

*then there exists $b_n < \infty$ such that for any $\psi_n \in B\left(\mathcal{X}^{n+1}\right)$*

$$\mathbb{E}\left[\left\{\widetilde{\beta}_n^{N_0}\left(\psi_n\right) - \alpha_n\left(\psi_n\right)\right\}^2\right] \leq \frac{b_n}{N_0}\left\|\psi_n\right\|^2. \tag{5}$$

**Proof.** We have

$$\widetilde{\beta}_n^{N_0}\left(\psi_n\right) - \alpha_n\left(\psi_n\right) = \widetilde{\beta}_n^{N_0}\left(\psi_n\right) - \beta_n^{N_0}\left(\psi_n\right) + \beta_n^{N_0}\left(\psi_n\right) - \alpha_n\left(\psi_n\right)$$

so by Minkowski's inequality

$$\mathbb{E}^{1/2}\left[\left\{\beta_n^{N_0}\left(\psi_n\right) - \alpha_n\left(\psi_n\right)\right\}^2\right] \leq \mathbb{E}^{1/2}\left[\left\{\widetilde{\beta}_n^{N_0}\left(\psi_n\right) - \beta_n^{N_0}\left(\psi_n\right)\right\}^2\right] + \mathbb{E}^{1/2}\left[\left\{\beta_n^{N_0}\left(\psi_n\right) - \alpha_n\left(\psi_n\right)\right\}^2\right].$$

The second term on the rhs is bounded using (4), so it suffices to control the first term. We have

$$\widetilde{\beta}_n^{N_0}\left(\psi_n\right) - \beta_n^{N_0}\left(\psi_n\right) = \frac{1}{N_0}\sum_{i=1}^{N_n}\left(M_{n+1}^i V_n^i - W_n^i\right)\psi_n\left(X_{0:n}^{i,n}\right)$$

$$\mathbb{E}\left[\left\{\widetilde{\beta}_n^{N_0}\left(\psi_n\right) - \beta_n^{N_0}\left(\psi_n\right)\right\}^2 \middle| \mathcal{F}_n\right] = \frac{1}{N_0^2}\mathbb{E}\left[\left\{\sum_{i=1}^{N_n}\left(M_{n+1}^i V_n^i - W_n^i\right)\psi_n\left(X_{0:n}^{i,n}\right)\right\}^2 \middle| \mathcal{F}_n\right]$$

where $M_{n+1}^i$ is the number of children of particle $i$ and $V_n^i$ their common weight. Using the specific structure of the branching step, these are independent across particles, so,

$$\mathbb{E}\left[\left\{\sum_{i=1}^{N_n}\left(M_{n+1}^i V_n^i - W_n^i\right)\psi_n\left(X_{0:n}^{i,n}\right)\right\}^2 \middle| \mathcal{F}_n\right]$$

$$= \sum_{i=1}^{N_n}\mathbb{E}\left[\left(M_{n+1}^i V_n^i - W_n^i\right)^2 \middle| \mathcal{F}_n\right]\psi_n\left(X_{0:n}^{i,n}\right)^2$$

$$\leq \sum_{i=1}^{N_n}\mathbb{V}\left[M_{n+1}^i V_n^i \middle| \mathcal{F}_n\right]\left\|\psi_n\right\|^2$$

Using Assumption V, Now $M_{n+1}^i$ is a translated Bernoulli variable and has variance upper bounded by $1/4$, so

$$\mathbb{E}\left[\left.\left\{\sum_{i=1}^{N_n}\left(M_{n+1}^i V_n^i - W_n^i\right)\psi_n\left(X_{0:n}^{i,n}\right)\right\}^2\right|\mathcal{F}_n\right] \leq \sum_{i=1}^{N_n} V\mathbb{E}\left[\left.(\overline{W}_n^i)^2\right|\mathcal{F}_n\right]\|\psi_n\|^2$$

Using $\overline{W}_n^i \leq 1$,
$$\leq \sum_{i=1}^{N_n} V\mathbb{E}\left[\left.\overline{W}_n^i\right|\mathcal{F}_n\right]\|\psi_n\|^2$$

Using Assumption O,
$$= \sum_{i=1}^{N_n} V\frac{1}{N_n}\sum_{i=1}^{N_n} W_n^i\|\psi_n\|^2$$

$$= V\sum_{i=1}^{N_n} W_i\|\psi_n\|^2.$$

Now it follows from the unbiasedness of the marginal likelihood estimate that

$$\mathbb{E}\left[\left\{\sum_{i=1}^{N_n}\left(M_{n+1}^i V_n^i - W_n^i\right)\psi_n\left(X_{0:n}^{i,n}\right)\right\}^2\right] \leq V\|\psi_n\|^2 N_0 p(y_{0:n}).$$

Hence, it follows that

$$\mathbb{E}\left[\left\{\widetilde{\beta}_n^{N_0}(\psi_n) - \beta_n^{N_0}(\psi_n)\right\}^2\right] \leq \frac{Vp(y_{0:n})}{N_0}\|\psi_n\|^2.$$

∎

**Proposition 5 (Extend Step)** *Assume that there exists $b_n < \infty$ such that for any $\psi_n \in B\left(\mathcal{X}^{n+1}\right)$*

$$\mathbb{E}\left[\left\{\widetilde{\beta}_n^{N_0}(\psi_n) - \alpha_n(\psi_n)\right\}^2\right] \leq \frac{b_n}{N_0}\|\psi_n\|^2 \tag{6}$$

*then there exists $c_n < \infty$ such that for any $\psi_{n+1} \in B\left(\mathcal{X}^{n+2}\right)$*

$$\mathbb{E}\left[\left\{\widehat{\beta}_{n+1}^{N_0}(\psi_{n+1}) - \widehat{\alpha}_{n+1}(\psi_{n+1})\right\}^2\right] \leq \frac{c_n}{N_0}\|\psi_{n+1}\|^2. \tag{7}$$

**Proof.** By Minkowski's inequality,

$$\mathbb{E}^{1/2}\left[\left\{\widehat{\beta}_{n+1}^{N_0}(\psi_{n+1}) - \widehat{\alpha}_{n+1}(\psi_{n+1})\right\}^2\right]$$

$$\leq \mathbb{E}^{1/2}\left[\left\{\widehat{\beta}_{n+1}^{N_0}(\psi_{n+1}) - \mathbb{E}[\widehat{\beta}_{n+1}^{N_0}(\psi_{n+1})|\widetilde{\mathcal{F}}_n]\right\}^2\right] + \mathbb{E}^{1/2}\left[\left\{\mathbb{E}[\widehat{\beta}_{n+1}^{N_0}(\psi_{n+1})|\widetilde{\mathcal{F}}_n] - \widehat{\alpha}_{n+1}(\psi_{n+1})\right\}^2\right]$$

The second term is,

$$\mathbb{E}^{1/2}\left[\left\{\mathbb{E}[\widehat{\beta}_{n+1}^{N_0}(\psi_{n+1})|\widetilde{\mathcal{F}}_n] - \widehat{\alpha}_{n+1}(\psi_{n+1})\right\}^2\right] = \mathbb{E}^{1/2}\left[\left\{\widetilde{\beta}_n^{N_0}(f_n(\psi_{n+1})) - \alpha_n(f_n(\psi_{n+1}))\right\}^2\right]$$

$$\leq \frac{b_n}{N_0}\|f_n(\psi_{n+1})\|^2 \leq \frac{b_n}{N_0}\|\psi_{n+1}\|^2$$

For the first term, we have,

$$\widehat{\beta}_{n+1}^{N_0}\left(\psi_{n+1}\right) - \mathbb{E}[\widehat{\beta}_{n+1}^{N_0}\left(\psi_{n+1}\right)|\widetilde{\mathcal{F}}_n]$$

$$=\frac{1}{N_0}\left\{\sum_{i=1}^{N_{n+1}}\widetilde{W}_n^i\psi_{n+1}\left(X_{0:n+1}^{i,n+1}\right) - \sum_{i=1}^{N_{n+1}}\widetilde{W}_n^i f_n\psi_{n+1}\left(X_{0:n}^{i,n+1}\right)\right\}$$

$$=\frac{1}{N_0}\sum_{i=1}^{N_{n+1}}\widetilde{W}_n^i\left(\psi_{n+1}\left(X_{0:n+1}^{i,n+1}\right) - f_n\psi_{n+1}\left(X_{0:n}^{i,n+1}\right)\right).$$

Hence by taking expectations,

$$\mathbb{E}\left[\left\{\widehat{\beta}_{n+1}^{N_0}\left(\psi_{n+1}\right) - \mathbb{E}[\widehat{\beta}_{n+1}\left(\psi_{n+1}\right)|\widetilde{\mathcal{F}}_n]\right\}^2\bigg|\widetilde{\mathcal{F}}_n\right]$$

$$=\frac{1}{N_0^2}\sum_{i=1}^{N_{n+1}}\left(\widetilde{W}_n^i\right)^2\mathbb{E}\left[\left(\psi_{n+1}\left(X_{0:n+1}^{i,n+1}\right) - f_n\psi_{n+1}\left(X_{0:n}^{i,n+1}\right)\right)^2\bigg|\widetilde{\mathcal{F}}_n\right]$$

$$\leq\frac{1}{N_0^2}\sum_{i=1}^{N_{n+1}}\widetilde{W}_n^i 2\|\psi_{n+1}\|^2$$

$$=\frac{2}{N_0^2}\sum_{i=1}^{N_n}W_n^i\|\psi_{n+1}\|^2$$

By unbiasedness of the marginal likelihood estimate,

$$\mathbb{E}\left[\left\{\widehat{\beta}_{n+1}^{N_0}\left(\psi_{n+1}\right) - \mathbb{E}[\widehat{\beta}_{n+1}\left(\psi_{n+1}\right)|\widetilde{\mathcal{F}}_n]\right\}^2\right] \leq \frac{2p(y_{0:n})}{N_0}\|\psi_{n+1}\|^2$$

∎

**Proposition 6 (Reweighing Step)** *Assume that there exists $c_n < \infty$ such that for any $\psi_{n+1} \in B\left(\mathcal{X}^{n+2}\right)$*

$$\mathbb{E}\left[\left\{\widehat{\beta}_{n+1}^{N_0}\left(\psi_{n+1}\right) - \widehat{\alpha}_{n+1}\left(\psi_{n+1}\right)\right\}^2\right] \leq \frac{c_n}{N_0}\|\psi_{n+1}\|^2 \qquad (8)$$

*then there exists $a_{n+1} < \infty$ such that for any $\psi_{n+1} \in B\left(\mathcal{X}^{n+2}\right)$*

$$\mathbb{E}\left[\left\{\beta_{n+1}^{N_0}\left(\psi_{n+1}\right) - \alpha_{n+1}\left(\psi_{n+1}\right)\right\}^2\right] \leq \frac{a_{n+1}}{N_0}\|\psi_{n+1}\|^2. \qquad (9)$$

**Proof.** We have

$$\beta_{n+1}^{N_0}\left(\psi_{n+1}\right) - \alpha_{n+1}\left(\psi_{n+1}\right) = \widehat{\beta}_{n+1}^{N_0}(g_{n+1}\psi_{n+1}) - \widehat{\alpha}_{n+1}\left(g_{n+1}\psi_{n+1}\right),$$

so

$$\mathbb{E}\left[\left\{\beta_{n+1}^{N_0}\left(\psi_{n+1}\right) - \alpha_{n+1}\left(\psi_{n+1}\right)\right\}^2\right] \leq \frac{c_n}{N_0}\|g_{n+1}\psi_{n+1}\|^2 \leq \frac{c_n}{N_0}\|\psi_{n+1}\|^2.$$

∎ The following Proposition shows that it is straightforward to transfer the L2 error bounds on $\beta_n^{N_0}$, $\widetilde{\beta}_n^{N_0}$ and $\widehat{\beta}_n^{N_0}$ to $\nu_n^{N_0}$, $\widetilde{\nu}_n^{N_0}$ and $\widehat{\nu}_n^{N_0}$.

**Proposition 7 (Normalisation)** *Assume we have an unnormalised random measure $\mu^{N_0}(dx) = N_0^{-1} \sum_{i=1}^{N} W_i \, \delta_{X^i} dx$ on $E$ where $0 < W_i \leq 1$ almost surely and such that there exists a measure $\mu$ and a constant $c < \infty$ satisfying for any $\psi \in B(E)$*

$$\mathbb{E}\left[\left\{\mu^{N_0}(\psi) - \mu(\psi)\right\}^2\right] \leq \frac{c}{N_0}\|\psi\|^2 \tag{10}$$

*then there exists a constant $\bar{c} < \infty$ such that for any $\psi \in B(E)$*

$$\mathbb{E}\left[\left\{\frac{\mu^{N_0}(\psi)}{\mu^{N_0}(1)} - \frac{\mu(\psi)}{\mu(1)}\right\}^2\right] \leq \frac{\bar{c}}{N_0}\|\psi\|^2.$$

**Proof.** We have

$$\frac{\mu^{N_0}(\psi)}{\mu^{N_0}(1)} - \frac{\mu(\psi)}{\mu(1)} = \frac{\mu^{N_0}(\psi)}{\mu^{N_0}(1)} - \frac{\mu^{N_0}(\psi)}{\mu(1)} + \frac{\mu^{N_0}(\psi)}{\mu(1)} - \frac{\mu(\psi)}{\mu(1)}$$

$$= \frac{\mu^{N_0}(\psi)\left\{\mu(1) - \mu^{N_0}(1)\right\}}{\mu^{N_0}(1)\,\mu(1)} + \frac{\mu^{N_0}(\psi) - \mu(\psi)}{\mu(1)}$$

so

$$\left|\frac{\mu^{N_0}(\psi)}{\mu^{N_0}(1)} - \frac{\mu(\psi)}{\mu(1)}\right| \leq \frac{\|\psi\|\left|\mu^{N_0}(1) - \mu(1)\right|}{\mu(1)} + \frac{\left|\mu^{N_0}(\psi) - \mu(\psi)\right|}{\mu(1)}.$$

Hence by Minkowski's inequality

$$\mathbb{E}^{1/2}\left[\left\{\frac{\mu^{N_0}(\psi)}{\mu^{N_0}(1)} - \frac{\mu(\psi)}{\mu(1)}\right\}^2\right] \leq \frac{\|\psi\|}{\mu(1)}\mathbb{E}^{1/2}\left[\left\{\mu^{N_0}(1) - \mu(1)\right\}^2\right]$$

$$+ \frac{1}{\mu(1)}\mathbb{E}^{1/2}\left[\left\{\mu^{N_0}(\psi) - \mu(\psi)\right\}^2\right]$$

and the result follows from (10). $\blacksquare$

The following Theorem now follows directly from the previous Proposition and Theorem on L2 error bounds for unnormalised measures.

**Theorem 8 (L2 error bounds for normalised measures)** *For any $n \geq 0$, there exists $\bar{a}_n, \bar{b}_n, \bar{c}_n < \infty$ such that for any $N_0 \geq 1$ and any $\psi_n \in B\left(\mathcal{X}^{n+1}\right)$, $\psi_{n+1} \in B\left(\mathcal{X}^{n+2}\right)$*

$$\mathbb{E}\left[\left\{\nu_n^{N_0}(\psi_n) - \eta_n(\psi_n)\right\}^2\right] \leq \frac{\bar{a}_n}{N_0}\|\psi_n\|^2,$$

$$\mathbb{E}\left[\left\{\widetilde{\nu}_n^{N_0}(\psi_n) - \eta_n(\psi_n)\right\}^2\right] \leq \frac{\bar{b}_n}{N_0}\|\psi_n\|^2,$$

$$\mathbb{E}\left[\left\{\widehat{\nu}_{n+1}^{N_0}(\psi_{n+1}) - \widehat{\eta}_{n+1}(\psi_{n+1})\right\}^2\right] \leq \frac{\bar{c}_n}{N_0}\|\psi_{n+1}\|^2.$$

# 4 Number of Particles

**Proposition 9** *The numbers of particles $(N_n)_{n \geq 0}$ is a martingale.*

**Proof.** We will show that $\mathbb{E}[N_{n+1}|\mathcal{F}_n] = N_n$ by showing that for each particle $i = 1, \ldots, N_n$, the expected number of children $\mathbb{E}[M_{n+1}^i|\mathcal{F}_n] = 1$. Using Assumption O, that the branching step involves a uniformly random ordering over particles,

$$\mathbb{E}[M_{n+1}^i|\mathcal{F}_n] = \mathbb{E}\left[\frac{W_n^{\sigma_n(i)}}{\overline{W}_n^i}\Big|\mathcal{F}_n\right]$$

$$= \mathbb{E}\left[\frac{1}{i}\sum_{j=1}^{i}\frac{W_n^{\sigma_n(j)}}{\overline{W}_n^i}\Big|\mathcal{F}_n\right]$$

$$= 1$$

since $\overline{W}_n^i = \frac{1}{i}\sum_{j=1}^{i} W_n^{\sigma_n(j)}$ and $\sigma_n$ is a uniform random permutation. ∎

**Proposition 10** *We have*

$$\mathbb{V}[N_n] \leq nVN_0$$

*for some constant $V$.*

**Proof.** We proof this by induction on $n$. The case $n = 0$ is trivial since $\mathbb{V}[N_0] = 0$. Recall that

$$N_{n+1} = \sum_{i=1}^{N_n} M_{n+1}^i$$

with $M_{n+1}^i$ being independent given $\mathcal{F}_n$, with variance $\mathbb{V}[M_{n+1}^i|\mathcal{F}_n] \leq V$ by Assumption V. Suppose the proposition is true for $n$. Then,

$$\mathbb{V}[N_{n+1}|\mathcal{F}_n] \leq VN_n$$
$$\mathbb{V}[N_{n+1}] = \mathbb{E}\left[\mathbb{V}[N_{n+1}|\mathcal{F}_n]\right] + \mathbb{V}\left[\mathbb{E}[N_{n+1}|\mathcal{F}_n]\right]$$
$$\leq \mathbb{E}[VN_n] + \mathbb{V}[N_n]$$
$$\leq (n+1)VN_0$$

∎

As a consequence, the standard deviation is $\sqrt{nVN_0}$. Then the standard deviation can be made arbitrarily small relative to the expected number of particles, $N_0$, by having $N_0$ arbitrarily larger than $Vn$.

**Corollary 11** *Using Doob's maximal inequality, we can also control the pathwise fluctuations of $(N_n)_{n \geq 0}$:*

$$\mathbb{E}[\sup_{k=1,\ldots,n}\left(\frac{N_k}{N_0} - 1\right)^2] \leq \frac{4n}{N_0}V = \frac{n}{N_0}.$$

Figure 1:   In this figure we demonstrate potential consequences when Assumption O is violated, comparing a best-case situation where the ordering of particles at $n$ is completely independent of the ordering of particles at $n + 1$, artificially subjecting the ordering of the particles to a random permutation, to a worst-case situation where the ordering of particles is completely preserved from $n$ to $n + 1$. We plot the number of particles $K_n$ at each of $n = 1, \ldots, 50$ for a one-dimensional linear Gaussian model, initialized with 100 particles. (left) When the order of the particles arriving at each $n$ is subject to a random permutation, then the number of particles is reasonably stable, staying at or near 100. (right) When the order of the particles arriving at each $n$ is completely deterministic, then the total number of particles quickly explodes, in this case exceeding 15000 by $n = 11$. In practice, a naïve implementation of the incremental resampling scheme will have a very strong dependence in ordering across $n$ — a particle which is one of the first to reach stage $n$ is quite likely one of the first to reach stage $n + 1$ as well.