[Reviews · NeurIPS 2014]

Submitted by Assigned_Reviewer_2

In particle filtering, the resampling step is a synchronous operation: one needs all the particles before computing the normalised weights (since the denominator is the sum of all the weights), and then resample. The reviewed paper propose an asynchronous resampling mechanism, where the number of children of particle k depends only the weights of particles 1 to k.
The proposed idea is quite straightforward, but still interesting and potentially very useful.
What is a bit lacking in the current version is some motivation for an asynchronous implementation of particle filtering. Having a different thread for each particles seems quite involved for many practical cases (recall that not every user of particle filtering works in CS), so providing a more substantial discussion (beyond citing two 2014 papers) would be tremendously helpful. Perhaps one step in this direction is to clarify a bit the discussion of Figure 3. Is it the case that "SMC" means a version of SMC where the set of particles are distributed over the k cores, and then "re-united" before each resampling step? Then the better performance of the particle casade would be a convincing argument. Another motivation would be to look at more realistic examples, in particular cases where the running time of simulating a single particle is very variable (such as in e.g. models in stochastic chemometrics, off the top of my head)

Two more points:
1. in most cases, people are interested in on-line algorithms: i.e. datapoints are collected sequentially, and as soon as y_n is available, computations may resume to deliver the filtering distribution at time n, etc. Do I understand correctly that the proposed algorithm is not on-line any more (see 305-306)? (If yes, then this point should be mentioned; also perhaps it's easy to make it on-line again?
2. Why did you compare particle casade to CSMC??? Sorry, I really did not understand this. CSMC samples from the smoothing distribution (if you take the selected trajectory), and is not really meant to be a filtering algorithm???

Minor remarks:
27: first sentence is a bit unclear; it's oddly constructed.
262: posterior marginals: do you mean filtering marginals (conditional on data up to time n), or smoothing marginals (conditional up to final time N)? I guess the former, but your terminology seems to mean the latter. Also does the MSE correspond to the filterting expectation of the state in Figures 1 and 2? See also 396 "true posterior distribution".
428-430: sorry, I did not understand this part.
Summary: A simple, elegant solution to the barrier synchronisation problem in particle filtering, but the problem itself should be a bit more motivated.

Submitted by Assigned_Reviewer_6

In this paper, the authors propose a new sequential Monte Carlo (SMC) technique for particle filtering which they call the particle cascade. The particle cascade differs from traditional SMC in that, instead of a resampling step, the particles are branched one at a time, with the number of offspring of a particle dependent only on those particles which have already been branched. The result is an algorithm which has comparable efficiency to standard SMC but avoids the synchronous resampling step, thereby providing more flexibility in implementation (e.g., easier parallelization, new particles can initialized during the run if necessary).

The work is properly motivated. The paper describes standard SMC and the issues associated with its synchronous resampling step, thereby demonstrating the utility of an anytime alternative. The limitations of existing anytime SMC methods (iterated conditional SMC) are discussed.

The work is also appropriately illustrated. Comparison of the performance (for two experiments) indicates that the particle cascade achieves similar results to that of traditional SMC and outperforms iterated conditional SMC. Due to these promising results and the fact the results are communicated using clear writing, I recommend acceptance.

I would like to see some discussion motivating the choice of the branching decision given by equation (9), over alternatives. An experiment showing the variability of K_n for different choices (similar to the experiment illustrated by Figure 1 of the supplement) could provide an argument for the current choice.

A related work that would be helpful to add in the literature review is:

Sequentially interacting Markov chain Monte Carlo methods
Anthony Brockwell, Pierre Del Moral, Arnaud Doucet

Some small notes:

-Line 150: While higher weights can be associated with high probability, it is more indicative of high posterior probability relative to the proposal q used, not necessarily absolutely.
-Line 200 In "analogously to before" is not immediately obvious what is meant by "before".
-Bibliography style seems non-standard for NIPS.
Summary: Comparison of the performance (for two experiments) indicates that the particle cascade achieves similar results to that of traditional SMC and outperforms iterated conditional SMC. Due to these promising results and the fact the results are communicated using clear writing, I recommend acceptance.

Submitted by Assigned_Reviewer_7

The problem addressed in the manuscript is that of the efficient implementation of the resampling step involved in the implementation of SMC methods. This is a difficult problem which has attracted attention for many years. The approach suggested in order to avoid full communication between samples is to introduce a form of sequential communication between samples at each time steps (the cascade) which offers some freedom in the implementation of this step. The key step is described in (9) and above, and is clearly described. The scheme preserves some important properties of standard implementations of SMC, which is nice and potentially useful in the context of pseudo-marginal algorithms.

However, and although I think that I have understood/guessed how the authors take advantage of this idea when implementing the algorithm (in particular on multi-core machines) the description in section 4 is somewhat unclear. I think that this section should be rewritten since it is crucial to link (9) and asynchronicity to parallel architectures. Perhaps the flow of successive operations on a toy architecture could be shown on a graph.

The simulations and comments, although brief, are convincing.
Summary: This is interesting and good work. The implementation of the algorithm is not always clearly described and could be improved.
Author Feedback
Author rebuttal: We thank all the reviewers for their thoughtful feedback. Responses to individual points below:

To Assigned_Reviewer_2:

> posterior marginals: do you mean filtering marginals (conditional on data up to time n), or smoothing marginals (conditional up to final time N)?

We will clarify that the particle cascade algorithm can be used for computing either the “filtering” distribution (conditioning on the first n data points) or “smoothing” distribution (conditioning on all N data points). All experiments in this paper are performed on the full smoothing distribution, which we refer to in general simply as the posterior marginals (e.g. at line 362).

Expectations computed as described at 200-202 are expectations over the posterior (smoothing) distribution. Expectations over filtering distribution at observation n could be computed by using the weights W_n, instead of the weights W_N.

> […] clarify a bit the discussion of Figure 3. Is it the case that "SMC" means a version of SMC where the set of particles are distributed over the k cores, and then "re-united" before each resampling step?

Yes — this is exactly what is meant by SMC in Figure 3. The example application you suggest ("… cases where the running time of simulating a single particle is very variable") is precisely a situation where the asynchronous resampling scheme is beneficial, as individual cores that ran fast simulations will not need to wait while the slower simulations complete.

> 1. in most cases, people are interested in on-line algorithms […]

While we do not discuss the details in the paper, it is straightforward to run the particle cascade in an online setting. We do this by simply keeping a pool of particles around once they have finished evaluating all the current data points; when a new data point is collected, we can then resume execution of these particles.

> 2. Why did you compare particle casade to CSMC? […] CSMC samples from the smoothing distribution […]

In the experiments section, we compare to CSMC because we are, in fact, using the particle cascade to sample from the full posterior (smoothing) distribution, conditioned on all Y_1, …, Y_N.

To Assigned_Reviewer_3:

> I would like to see some discussion motivating the choice of the branching decision given by equation (9), over alternatives.

One motivation for the branching decision in equation 9 can be found by considering the situation where the deterministic floor / ceiling rule for the R > 1 case is replaced by a Bernoulli flip, and the estimated average weight is approximately the true average weight (which holds true when the number of particles is large). Then, this decision rule defines the same distribution on the number of offspring particles sampled at each n as the well-known (and well-performing) systematic resampling procedure. We will add details motivating this choice to the final paper.

To Assigned_Reviewer_6:

> although I think that I have understood […] the description in section 4 is somewhat unclear […] it is crucial to link (9) and asynchronicity to parallel architectures.

Thanks for the feedback and suggestions; we will edit section 4 for clarity for the final version of the paper.

> A related work that would be helpful to add in the literature review […]

Thanks; we will add this to the literature review.